# Trajectories of Meat Intake and Risk of Type 2 Diabetes: Findings from the China Health and Nutrition Survey (1997–2018)

**DOI:** 10.3390/nu15143277

**Published:** 2023-07-24

**Authors:** Mengran Liu, Huijun Wang, Shufa Du, Yingying Jiao, Qi Wang, Chang Su, Bing Zhang, Gangqiang Ding

**Affiliations:** 1Department of Education and Training, Chinese Center for Disease Control and Prevention, Beijing 102206, China; liumr@chinacdc.cn; 2Key Laboratory of Trace Element Nutrition of National Health Commission of China, National Institute for Nutrition and Health, Chinese Center for Disease Control and Prevention, Beijing 100050, China; 3Department of Nutrition and Carolina Population Center, University of North Carolina at Chapel Hill, Chapel Hill, NC 27599, USA

**Keywords:** Chinese adults, dose–response relationship, meat intake, type 2 diabetes, trajectories

## Abstract

Few articles have investigated the impact of long-term meat intake trends and their changes during follow-up on the risk of type 2 diabetes (T2D). We aimed to explore the long-term trajectories of meat intake and determine its association with T2D risk in Chinese adults. This study used seven rounds of data from the China Health and Nutrition Survey (1997, 2000, 2004, 2006, 2009, 2015, and 2018), and 4464 adults aged 18 years or older were analyzed. The group-based trajectory modeling was used to identify meat intake trajectories over 21 years. Multivariate Cox proportional hazard and restricted cubic spline models were used to analyze the association and dose–response relationship between meat intake and T2D. Four trajectory groups were identified: “low-increase intake group” (Group 1), “moderate-increase intake group” (Group 2), “medium-stable intake group” (Group 3), and “high intake group” (Group 4). Compared with Group 2, Group 4 was associated with a higher risk of developing T2D (hazard ratio 2.37 [95% CI 1.41–3.98]). After adjusting for demographic characteristics, lifestyle, total energy intake, waist circumference, and systolic blood pressure, and using the third quintile as a reference, the risk of T2D was increased by 46% in the lowest quintile with meat intake (hazard ratio 1.46 [95% CI 1.07–2.01]) and by 41% in the highest quintile with meat intake (HR 1.41 [95% CI 1.03–1.94]). A U-shape was observed between meat intake and T2D risk (*p* for nonlinear < 0.001). When the intake was lower than 75 g/day, the risk of T2D was negatively correlated with meat intake, while the risk of T2D was positively correlated with meat intake when the intake was higher than 165 g/day. We identified four trajectory groups of meat intake from 1997 to 2018, which were associated with different risks of developing T2D. A U-shaped association was observed between meat intake and T2D in Chinese adults.

## 1. Introduction

The prevention and control of diabetes is an important public health challenge worthy of attention in recent years. According to the International Diabetes Federation’s Diabetes Atlas, in 2021 the number of patients aged 20–79 with diabetes was estimated to be 536 million, which is predicted to be 783 million by 2045 [1]. Existing evidence suggests that a sedentary lifestyle, suboptimal diet, lack of exercise, and excessive obesity are the main risk factors for developing T2D [2]. Efforts should be made to reduce the incidence rate, complications, and mortality of diabetes.

Over the past few decades, there has been a lot of consistent evidence showing the importance of dietary factors in the prevention of T2D, in observational study and clinical trials [3]. For example, there is a large amount of evidence that red meat, processed meat, sugar-sweetened beverages (SSBs), and western dietary patterns, characterized by high meat consumption, are positively related to the risk of diabetes, while whole grains and Mediterranean diet, characterized by high plant foods consumption, are associated with a lower risk of T2D [4,5,6]. Many studies have focused on the relationship between animal protein intake, total meat intake, red meat intake, poultry intake, processed meat intake, fish intake, and the risk of T2D [7,8,9,10]. At present, more evidence shows that red meat and processed meat are positively related to the risk of T2D, and there are also inconsistent results in some studies [11]. In addition, the evidence for a clear overall association of either poultry or fish intake with T2D risk is not clear enough. It is critical to consider the source of dietary protein and the subtype of meat in the study of dietary factors for preventing T2D.

In China, processed meat intake and red meat intake were responsible for 2.8 million and 1.8 million cases of diabetes in 2011, respectively [12]. Several studies have assessed the association of meat intake with the risk of T2D among Chinese adults, indicating that meat consumption is a related factor [13,14,15]. However, these studies are cross-sectional studies or provincial observational study. Although some cohort studies examined the association between meat consumption and T2D risk [16], most of these studies used a single time point as the assessment point of dietary exposure, or the dietary data at each round of follow-up are either partially obtained or unavailable. Few articles have investigated the impact of the long-term meat intake trend and its changes during follow-up, and also have not considered population heterogeneity. Based on the information we currently have, there is no research on the use of long-term intake trajectories of food groups as exposure measurements; that is, taking meat intake trajectories as latent variables.

The group-based trajectory model approach can identify individuals following similar trajectories of a single variable, and a comprehensive representation of the long-term exposure can be provided by the trajectory group [17]. Moreover, Chinese people tend to consume pork, followed by chicken, beef, etc., and the consumption of processed meat is not high among the overall Chinese population. Therefore, this study gave priority to explore the long-term trajectories of total meat intake and determine their association and dose–response relationship with T2D risk in Chinese adults using data from the CHNS. Such findings would be valuable for providing a scientific basis for developing and implementing public health actions to prevent and control T2D.

## 2. Materials and Methods

### 2.1. Study Design

We used data derived from seven waves of the CHNS (1997, 2000, 2004, 2006, 2009, 2015, and 2018), a longitudinal study initiated in 1989, and 10 rounds of follow-up were conducted from 1991 to 2018. More details regarding CHNS have been previously published [18].

To explore the long-term trajectories of meat intake, we identified the meat intake trajectories from 1997 to 2018, which we called the “track identification period”. Considering that dietary changes after developing T2D may confound the relationship between meat intake and the disease, we used only dietary information before diagnosis with T2D in prospective analyses.

Since we only collected fasting blood samples in 2009, 2015, and 2018, our cohort analysis on the relationship between meat intake and risk of T2D was based on these three rounds of data. We call the period from 2009 to 2018 “survival analysis period”.

### 2.2. Study Participants

For this study, we selected subjects who were adults aged 18 years or older, who were not pregnant or lactating, and who had participated in the 2009, 2015, and 2018 surveys (*n* = 24,372). We then excluded those who had a dietary data deficiency or abnormal daily energy intake, those who had a blood sample deficiency, and those who had a demographic information deficiency (see Figure 1).

Among the remainder (*n* = 19,499), participants in at least two follow-up waves from 2009 to 2018 who were not diagnosed with T2D when they first entered the “survival analysis period” were selected (*n* = 8886). From that group, participants who had fewer than three waves of available dietary data before diagnosis with T2D were excluded. Finally, we excluded those who self-reported diabetes in the 1997–2006 survey to ensure that all participants were nondiabetic when they entered the “survival analysis period”. A total of 4464 people were included in the analysis, finally.

The survey was approved by the institutional review committees of the National Institute for Nutrition and Health, Chinese Center for Disease Control and Prevention (No. 201524), and the informed consent forms were signed by all participants.

### 2.3. Measurement of Meat Intake

In the CHNS, data on food consumption were collected by three consecutive days of 24 h dietary recalls, including two weekdays and one weekend day. Three days of oil and condiments were collected and allocated to individuals according to the proportion of personal energy consumption in the household. The average daily intake of various foods and nutrients per person and the average of total energy intake (TEI) were calculated based on the China Food Composition Table [19].

For this study, we have defined “total meat” as total fresh meat. Meat includes muscle and organ meat from pork, beef, lamb, and poultry that had not been treated. This study used dietary data from 1997 to 2018 for participants who met the inclusion and exclusion criteria of the study design, to calculate the cumulative average of total meat intake and analyze the correlation and dose–response relationship between meat intake and T2D. For example, if a participant had four rounds of dietary data for 1997, 2004, 2009, and 2015, the cumulative average of the respondent’s total meat intake = (1997 meat intake + 2004 meat intake + 2009 meat intake + 2015 meat intake)/4.

### 2.4. Diagnostic Criteria of T2D

Fasting blood samples were collected via venipuncture, and glucose test were performed immediately with strict quality control. According to the diagnosis and classification of diabetes mellitus provisional report and the guidelines for the prevention and treatment of diabetes mellitus in China [20,21], T2D is defined as HbA1c ≥ 48.0 mmol/mol (6.5%) or/and FPG ≥ 7.0 mmol/L or/and treatments for diabetes, including oral hypoglycemic medication or insulin injections.

### 2.5. Assessment of Covariates

Our study assessed covariates between 2009 and 2018 (“survival analysis period”), and both baseline and follow-up covariates were used in the present study.

The following measures were considered covariates: gender; age (in years); education level (primary school and below, completed middle school, high school and above); per capita family income (tertiles: low, medium, high); geographic region (rural, urban); smoking and alcohol-drinking status (current smoker/drinker, nonsmoker/nondrinker); sleep duration (<6 h, 6–9 h, >9 h); total physical activity (METs h/week); chronic disease history (previous diagnosis with hypertension, myocardial infarction, apoplexy, or cancer); waist circumference (WC); and systolic blood pressure (SBP). We also considered other dietary related factors as adjusted covariates, including total energy intake (TEI) and baseline meat intake.

### 2.6. Statistical Analysis

Using group-based trajectory modeling to identify meat intake trajectories from 1997 to 2018, we determine the number of groups based on some criteria, including their interpretability, the probability of average posterior value members in each group being greater than 0.7, and the inclusion of at least 2% of the sample population in each group, as well as better model fit with lower Bayesian information criterion [22].

Baseline characteristics of the subjects were analyzed by Wilcoxon rank-sum (continuous variables) and Chi-squared (categorical variables), according to estimated latent trajectory groups of meat intake. A two-level Cox proportional hazards regression model analysis was conducted (one level for individuals and two levels for families) to examine the associations between meat intake and the risk of T2D. We constructed four sequential models that included demographic characteristics, lifestyle, WC, SBP, and other dietary factors. Finally, the dose–response relationship between meat intake and T2D was analyzed using the restricted cubic spline (RCS) model.

All the analyses were conducted in SAS (Version 9.4, SAS Institute, Cary, NC, USA), Stata/SE (Version 15.0, StataCorp, Station, TX, USA), and R software (Version 4.1.0, R Foundation for Statistical Computing, Vienna, Austria,). *p* < 0.05 is defined as statistical significance.

## 3. Results

### 3.1. Trajectory Groups of Meat Intake

Figure 2 shows four trajectory groups of meat intake that we identified from 1997 to 2018. The first group comprised 19.8% of the participants; this group had the lowest initial meat intake and a slight increase thereafter and was thus labeled the “low-increase intake group”. The second trajectory group comprised 46.3% of the participants and was characterized by moderate meat intake, about 45–85 g/day, so this group was labeled the “moderate-increase intake group”. The third trajectory group comprised 31.0% of the participants whose meat intake was over 100 g/day and remained above average. Therefore, this group was labeled the “medium-stable intake group”. The participants in the fourth trajectory group initially had an above-average meat intake, with a slight increase and then a slight decrease, characterized by about 200–300 g/day of meat intake. Therefore, this group was labeled the “high intake group”.

### 3.2. Baseline Characteristics by Trajectory Groups

Table 1 presents the baseline characteristics of different groups of participants in the analysis by meat intake trajectories. We observed significant differences in gender, age, geographic region, education level, individual income, smoking status, drinking status, and physical activity among trajectory groups (*p* < 0.01). Baseline WC, SBP, and diastolic blood pressure also differed among groups (*p* < 0.05).

In the baseline of follow-up between 1997 and 2018, participants in Group 4 had a higher baseline mean level of urbanicity score, TEI, and WC but lower physical activity. Group 4 had the highest proportion of males and was characterized by high education level, urban residence, high income, and current smoking and alcohol-drinking status. It also had the lowest proportion of people over the age of 65.

### 3.3. Trajectory Groups of Meat Intake and T2D

In our study, gender and meat intake interaction analyses were performed first. The results showed that there was no statistical significance (*p* = 0.26). Gender was further adjusted in the model as a confounding factor.

Among the 4644 participants without T2D at baseline, 442 developed T2D with an incidence rate of 9.90%. The average follow-up year is 7.54 years, and the total number of people with T2D is 33,659. The Cox multiple regression analysis results of the associations between meat intake trajectory groups and risk of T2D are shown in Table 2. We used Group 2 as the reference group because the Chinese Dietary Guidelines suggest a recommended meat intake for adults of 280–525 g per week [23]. The risk of T2D in the “high intake group” (Group 4) was more than twice that of the reference group (HR = 2.37, 95% CI = 1.41, 3.98) when adjusted for all covariates (age, gender, urban/rural residence, education level, income, cigarette smoking, alcohol consumption, physical activity, sleep time, disease history, baseline meat intake, TEI, WC, and SBP).

### 3.4. Cumulative Averages of Meat Intake and T2D

Table 3 presents the association between cumulative averages of meat intake and risk of T2D. Using the third quintile as a reference, the risk of T2D was increased by 46% in the lowest quintile with meat intake (HR = 1.46, 95% CI = 2.01, 1.89) and by 41% in the highest quintile with meat intake (HR = 1.41, 95% CI = 1.03, 1.94) after adjustment for all covariates.

### 3.5. Dose–Response Relationship between Meat Intake and T2D

RCS analysis showed a U-shaped association between dietary meat intake and T2D (Figure 3). The overall and nonlinear associations between meat intake and the risk of T2D were statistically significant (*p* < 0.001). Using a meat intake of 75 g/day as a reference, when the meat intake was <75 g/day, the risk of T2D decreased significantly with the increase in meat intake. When the intake was between 75 and 165 g/day, meat intake was not statistically associated with T2D. The risk of T2D increased significantly with the increase in meat intake over 165 g/day.

## 4. Discussion

Four different meat intake trajectories were identified in this Chinese adult cohort study. New insights are provided into identifying populations at higher risk of T2D and the prevention of T2D. The research results also emphasize the importance of the long-term trajectory of meat intake on the subsequent risk of developing T2D, which is consistent with studies on the prevention and control of non communicable diseases during the life-course [24].

The etiology of T2D is associated with irreversible risk factors such as genetics, age, race, and ethnicity, as well as with reversible factors like diet, physical activity, and smoking, which are complex [25]. Many studies have shown that meat intake is associated with the risk of diabetes [26]. The follow-up study by the Health Professionals and the Nurses’ Health Study in the United States showed a positive correlation between the risk of developing T2D and the consumption of red meat. The pooled HRs in total, unprocessed, and processed red meat consumption for a one-serving-per-day increase were 1.14, 1.12, and 1.32, respectively. An increase of 42 g or more per day compounded the risk of diabetes over four years by 48% (pooled HR 1.48), comparing to no increase [27]. There is a significant correlation between new cases of diabetes and red meat consumption, which is also found in EPIC InterAct research [28]. One meta-analysis aims to quantitatively examine possible associations between intake of different types of meat and the risk of T2D. The summary relative risk of T2D was 1.33 for total meat intake, 1.22 for red meat, 1.25 for processed meat, and 1.00 for poultry [29]. No significant correlation was observed between meat consumption and T2D risk in some studies, which, in our opinion, may be related to the amounts and types of meat consumed in different world regions. Definitions used for selected meats may vary among individual studies, resulting in varying results. Over the past 20 years, China has undergone the dietary transition. The proportion consuming the minimum recommended amounts of whole grains and vegetables shows a significant decreasing trend, while the proportion exceeding the maximum recommended calories from meat shows an increasing trend [30]. Meat consumption patterns among Chinese people are different from that of in the west. Pork is the main type of meat consumption, followed by other meats and poultry. Based on the current high rate of total meat consumption among Chinese people, this study focuses on the associations between total meat intake and the risk of T2D.

The identification of food group trajectories focuses not only on the food group intake at the starting point of the trajectory but also on the long-term trend changes. We found that meat intake trajectories correlated with T2D. Compared with the “moderate-increase intake group”, T2D increased in the “high intake group”. In the “high intake group”, participants followed the highest initial meat intake and sustained a high intake trajectory of the meat intake, characterized by about 200–300 g/day meat intake during the follow-up. The increased risk of developing T2D may be related to long-term higher adherence to high meat consumption, as shown in this study. We did not observe this association in the “medium-stable intake group”, which was stable at about 130 g/day. That may be related to the inclusion of white meat in the total meat. In addition, the RCS model of this study shows no significant association for a meat intake of 75–165 g/day.

Meat is a valuable source of macro- and micronutrients, particularly proteins, vitamins A and B1, niacin, iron, and zinc. Either a very low intake or a larger intake of meat can have adverse health effects. A U-shaped relationship was found for dietary meat intake and risk of developing T2D. When the meat intake was <75 g/day or >165 g/day, the risk of T2D increased and reached statistical significance. When the meat intake was between 75 and 165 g/day, it was not statistically associated with T2D. The results of this study are consistent with the recommended intake of meat in the Chinese Dietary Guidelines, which suggest a recommended meat intake for adults of 280–525 g/week. We have reviewed relevant research and found that there is not much support in the literature for the explanation of the left branch of the U-shaped curve; that is, the impact of very low meat intake on the risk of T2D. We speculate that people with very low meat intake tend to shift towards high carbohydrate or fatty foods in order to maintain their energy needs, taste preferences, and satiety. This may be the possible reason for the increased risk of T2D. Highly refined carbohydrate foods, high-fat foods, and their combinations (sugar–oil mixtures) are associated with the risk of T2D.

We used two methods to evaluate dietary exposure when exploring the association between meat intake and risk of T2D. Trajectories of meat intake and cumulative average values of meat intake were examined for their relationship with the subsequent risk of T2D separately. In our results, the risk of T2D was increased in the lowest quintile with meat intake while no effects were observed in the “low-increase intake group”. This may be due to the lowest initial meat intake and a slight increase thereafter that reached 40 g/day. The increase in meat intake in the later stage offsets the potential risks associated with the extremely low initial intake.

The reasons for the adverse effects of meat in the development of diabetes have not been clarified, and it is usually assumed that there are several factors that act individually or in combination. An oversupply of heme iron due to a large intake of red meat is considered as a potential cause of atherosclerosis and diabetes [31]. Nitrite contained in processed meat will produce peroxynitrite during digestion, which may promote atherosclerosis and enhance the development of diabetes [32].

China has been in a period of nutritional transformation over recent decades. With the westernization of the diet pattern, the consumption of vegetables and whole grains has decreased, and the consumption of meat, processed meat and SSBs has increased [33]. It is estimated that poor dietary quality is significantly associated with a substantial proportion of mortality from stroke, heart disease, and T2D in China [30]. It is essential to understand the health outcomes of different food and nutrient intake trajectories over time. More detailed meat classification should be conducted, and its impact on the risk of T2D should be stratified by gender.

This study is the first to identify the long-term trajectory of meat intake in Chinese adults over 21 consecutive years, which breaks new ground by exploring trajectories of meat intake and their associations with T2D risk. However, our study also has some limitations. Firstly, trajectory recognition requires at least three rounds of data. We have imposed strict inclusion and exclusion criteria on our research design, which may reduce the sample size and the representativeness of the analyzed sample. Second, there may be some unmeasured confounding factors in this study, although as many potential covariates as possible have been adjusted by us. Thirdly, we used three 24 h dietary survey methods to evaluate meat and energy intake, which had some recall bias.

## 5. Conclusions

Four trajectory groups of meat intake identified were associated with different risks of developing T2D in Chinese adults, with the “high intake group”, characterized by people who consistently consume a large amount of meats, associated with increased risk of T2D. An overall U-shaped association between meat intake and risk of T2D is suggested in our research. Meat intake of more than 165 g/day might be a threshold point for developing T2D. These findings have important implications for preventing T2D, especially for people experiencing a transition in diet. It is necessary and meaningful to do research to verify the possible mechanism of association between meat intake and T2D.

## Figures and Tables

**Figure 1 nutrients-15-03277-f001:**
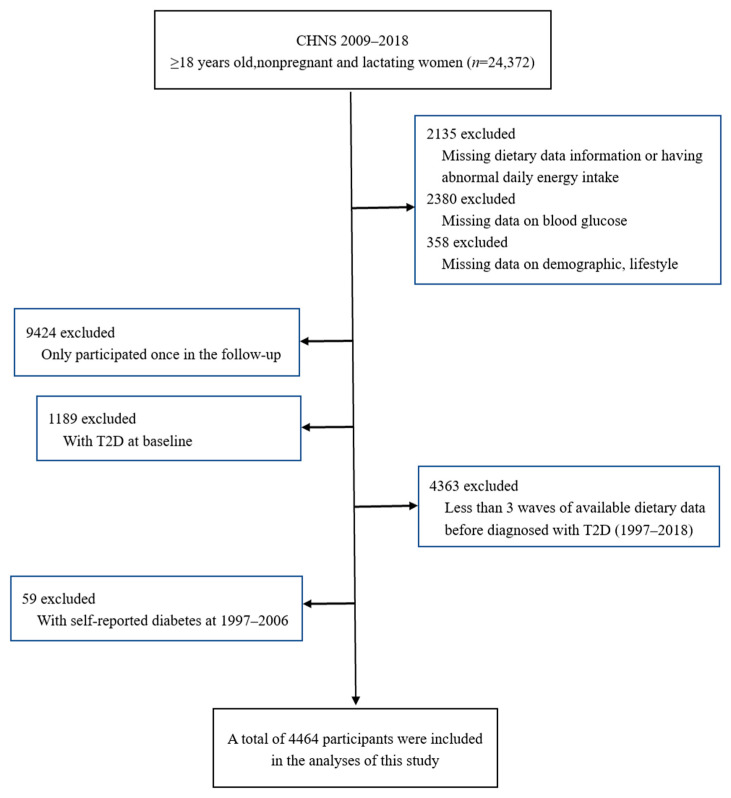
Flowchart of participants included in the analysis.

**Figure 2 nutrients-15-03277-f002:**
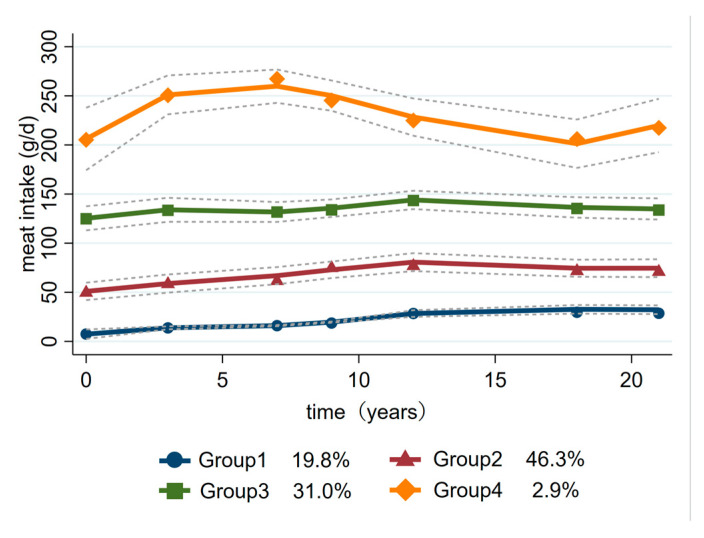
Estimated trajectory groups of meat intake among Chinese adults.

**Figure 3 nutrients-15-03277-f003:**
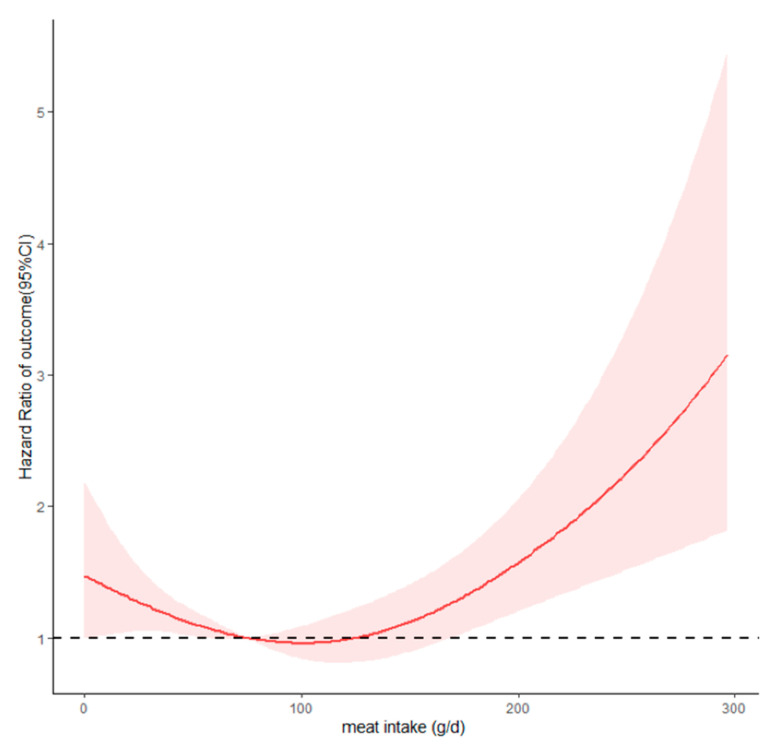
Dose–response relationship of dietary meat intake with type 2 diabetes.

**Table 1 nutrients-15-03277-t001:** Baseline characteristics by the four estimated latent trajectory groups.

Baseline Characteristics	Group 1	Group 2	Group 3	Group 4	*p* Value
	(*n* = 883)	(*n* = 2066)	(*n* = 1384)	(*n* = 131)	
Diabetes incidence rate (%)	10.19	9.39	8.45	16.03	0.032
Gender (%)					
Male	40.20	40.71	54.48	71.76	<0.001
Female	59.80	59.29	45.52	28.24	
Age groups (%)					
18–44 years	57.19	57.99	66.55	77.1	<0.001
45–64 years	39.75	38.38	31.14	20.61	
≥65 years	3.06	3.63	2.31	2.29	
Geographic region (%)					
Urban	9.51	24.78	37.86	45.8	<0.001
Rural	90.49	75.22	62.14	54.2	
Education level (%)					
Primary school	65.23	47.10	35.40	28.24	<0.001
Middle school	26.50	34.27	37.21	41.98	
High school and above	8.27	18.64	27.38	29.77	
Household income per capita (%)					
Low	55.72	32.33	21.75	20.61	<0.001
Median	29.22	35.04	33.60	31.30	
High	15.06	32.62	44.65	48.09	
Smoking (%)					
Nonsmoker	69.31	72.85	66.18	48.09	<0.001
Current smoker	30.69	27.15	33.82	51.91	
Alcohol drinking (%)					
Nondrinker	67.16	66.26	58.67	45.80	<0.001
Current drinker	32.84	33.74	41.33	54.20	
Physical activity (MET h/week)	423.17 ± 237.52	316.82 ± 254.31	266.13 ± 229.99	189.89 ± 162.84	<0.001
Urbanicity score (mean [SD])	46.59 ± 10.83	56.72 ± 12.57	60.80 ± 11.62	63.33 ± 11.86	<0.001
Total energy intake (kcal, mean [SD])	2462.88 ± 721.58	2377.75 ± 707.98	2537.40 ± 687.79	2771.13 ± 688.70	<0.001
BMI (mg/kg, mean [SD])	22.54 ± 3.05	22.7 ± 3.09	22.42 ± 2.93	22.64 ± 3.15	0.136
WC (cm, mean [SD])	78.74 ± 8.82	78.59 ± 9.75	77.62 ± 8.89	79.83 ± 10.06	0.004
SBP (mmHg, mean [SD])	119.35 ± 16.84	117.93 ± 16.36	116.28 ± 14.62	116.18 ± 13.40	<0.001
DBP (mmHg, mean [SD])	77.58 ± 11.03	77.08 ± 10.97	75.90 ± 9.76	77.41 ± 10.00	<0.001

Wilcoxon rank-sum and Kruskal–Wallis H tests were used for non-normally distributed continuous variables, and Chi-squared test was used for categorical variables.

**Table 2 nutrients-15-03277-t002:** Association between trajectory groups and risk of type 2 diabetes (*n* = 4464).

Baseline Characteristics	*n*	Cumulative Number of Cases/Person-Year	Model 1	Model 2	Model 3	Model 4
			Hazard Ratio (95% CI)	Hazard Ratio (95% CI)	Hazard Ratio (95% CI)	Hazard Ratio (95% CI)
Group 2	2066	194/15,619	1	1	1	1
Group 1	833	90/6249	1.10 (0.86, 1.42)	1.14 (0.88, 1.48)	1.16 (0.89, 1.50)	1.02 (0.78, 1.33)
Group 3	1384	117/10,477	0.89 (0.71, 1.12)	0.88 (0.70, 1.11)	0.88 (0.70, 1.12)	1.01 (0.78, 1.32)
Group 4	131	21/942	1.87 (1.19, 2.96) **	1.81 (1.14, 2.88) **	1.85 (1.16, 2.94) **	2.37 (1.41, 3.98) **

Model 1 adjusted for no covariates. Model 2 adjusted for age, sex, urban/rural, education level, and income. In Model 3, smoking, drinking, and physical activity levels, sleep duration, and disease history were further adjusted based on Model 2. In Model 4, total energy intake, waist circumference, systolic blood pressure, and baseline meat intake were further adjusted based on Model 3. ** *p* < 0.01.

**Table 3 nutrients-15-03277-t003:** Hazard ratio and 95% confidence interval for association of meat intake with type 2 diabetes.

	Quintile of Meat Intake (g/day)				
	Q1	Q2	Q3	Q4	Q5
	*n* = 892	*n* = 893	*n* = 893	*n* = 893	*n* = 893
Median (g/day)	13.97	45.28	75.45	112.78	166.25
Model 1	1.47 (1.09, 2.00) *	1.25 (0.91, 1.71)	1.00 (ref)	0.96 (0.69, 1.34)	1.29 (0.95, 1.76)
Model 2	1.54 (1.12, 2.10) *	1.29 (0.94, 1.77)	1.00 (ref)	0.95 (0.68, 1.32)	1.27 (0.93, 1.74)
Model 3	1.57 (1.15, 2.15) *	1.31 (0.96, 1.80)	1.00 (ref)	0.95 (0.68, 1.32)	1.29 (0.95, 1.77)
Model 4	1.46 (1.07, 2.01) *	1.35 (0.98, 1.85)	1.00 (ref)	1.00 (0.72, 1.40)	1.41 (1.03, 1.94) *

Model 1 adjusted for no covariates. Model 2 adjusted for age, sex, urban/rural, education level, and income. In Model 3, smoking, drinking, and physical activity levels, sleep duration, and disease history were further adjusted based on Model 2. In Model 4, total energy intake, waist circumference, and systolic blood pressure were further adjusted based on Model 3. * *p* < 0.05.

## Data Availability

The datasets generated and analyzed during the current study are available from the corresponding author (G.D.) upon reasonable request.

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
