# Peer review of "Trajectories of Meat Intake and Risk of Type 2 Diabetes: Findings from the China Health and Nutrition Survey (1997–2018)"

_nutrients, 2023, doi:10.3390/nu15143277_

Round 1
Reviewer 1 Report
Interesting study. Well presented. The methodology is appropriate to the research question.
I only have some minor comments:
Introduction:
Novelty of the study needs to be considered in the context of other studies from China: e.g. Huandong et al, Diabetologia 2020.
Meta-analysis from reference 6 which suggest that red meat increases risk of type 2 diabetes, needs to be balanced with the suggestion that in fact processed meat and not red meat increases risk of CVD and type 2 diabetes. (Micha et al, AHA 2020)
Methods:
Did you exclude participants with pre-diabetes? change in diet as a result of this diagnosis maybe a confounder if included. Did they exclude use of metformin for 'non-diabetes' indication (e.g. PCOS) or weight loss drugs.
For the measurement of meat intake, were investigators able to dissect between processed meat vs red meat vs total meat intake? Energy intake was not listed as covariate.
Others.
Methods or discussion section should discuss rationale for using group based trajectory modelling as opposed to other latent class modelling approaches. e.g. level of heterogenity within the groups.
Please speculate the reasons for relationship between reduce physical activity level with with trajectory for protein intake.
Limitations should also discuss non availability of data for type of meat, or used of processed meat as a confounder.
Author Response
Point 1: Novelty of the study needs to be considered in the context of other studies from China: e.g. Huandong et al, Diabetologia 2020.
Response 1:
Thank you for your suggestion. We have included other studies from China in the introduction section and discussed the novelty of the study.
Point 2: Meta-analysis from reference 6 which suggest that red meat increases risk of type 2 diabetes, needs to be balanced with the suggestion that in fact processed meat and not red meat increases risk of CVD and type 2 diabetes. (Micha et al, AHA 2020)
Response 2:
In the introduction, we cited many studies focusing on the relationship between animal protein, meat, red meat, processed meat, poultry, etc. and the risk of T2D. Their conclusions are not completely consistent. It is critical to consider the source of dietary protein and the subtype of meat in the study of dietary factors for preventing T2D.
Point 3: Did you exclude participants with pre-diabetes? change in diet as a result of this diagnosis maybe a confounder if included. Did they exclude use of metformin for 'non-diabetes' indication (e.g. PCOS) or weight loss drugs.
Response 3:
Our study takes type 2 diabetes as an outcome event. For people with pre-diabetes, if they are not within the scope of our diagnostic criteria, they will not be excluded. we ensured that all participants were nondiabetic when they entered the “survival analysis period”(2009-2018). From 2009 to 2018, the participants were tested for fasting blood samples to determine whether they had diabetes. From 1997 to 2006, diabetes was excluded in the form of self-report. Questionnaire was conducted to investigate whether diabetes was diagnosed, whether treatments for diabetes, including oral hypoglycemic medication or insulin injections.
Point 4: For the measurement of meat intake, were investigators able to dissect between processed meat vs red meat vs total meat intake? Energy intake was not listed as covariate.
Response 4:
Thank you for your suggestion. We have considered energy intake (TEI) as adjusted covariate. meat consumption patterns among Chinese are different from that of in the west. Chinese people tend to consume pork, followed by beef, chicken, etc., and the consumption of processed meat is not high among the overall Chinese population. Based on the current high rate of total meat consumption among Chinese people, this study give priority to focuses on the associations between total meat intake and the risk of T2D. Processed meat is not within the scope of our research, and there will be other studies (our research team) involving it.
Point 5: Methods or discussion section should discuss rationale for using group based trajectory modelling as opposed to other latent class modelling approaches. e.g. level of heterogenity within the groups.
Response 5:
The group-based trajectory model is an approach to capturing individual-level heterogeneity, which has been used in the field of nutritional epidemiology. The objective of the model is to identify distinct clusters of individuals by following similar patterns of an outcome measured over time and to profile the characteristics of group members.
Our research team has previously used this method to identify BMI trajectories, waist circumference trajectories, trajectory groups of daily energy intake distribution, trajectories of macronutrients intake, and has published articles. The group-based trajectory model is also used by many researchers in the fields of sociology and psychology. It is appropriate for us to use this method in this study, and we have attached two references to further enhance readers' understanding of this method. Considering that this method has been relatively mature in application, we did not provide a detailed introduction in the method section, but only used it as a statistical method tool.
Point 6: Please speculate the reasons for relationship between reduce physical activity level with with trajectory for protein intake.
Response 6:
We included four physical activity domains—occupational, household, leisure time, and transportation. Lack of occupational activity is the main factor contributing to low overall physical activity levels. We speculate that the population with fewer occupational activities, which the possibility of engaging in office and senior clerical work, generally have higher incomes and better food quality, thus observing a higher protein intake.
Point 7: Limitations should also discuss non availability of data for type of meat, or used of processed meat as a confounder.
Response 7:
A large number of studies have been conducted on the relationship between total meat intake, red meat intake, processed meat intake and the risk of diabetes, and the conclusions are inconsistent. This study focused on the relationship between total fresh meat intake and the risk of diabetes. Processed meat is not within the scope of our research, but it is indeed worth paying attention to and researching, and there will be other studies (our research team) involving it.
Thank you for your suggestions, which are very constructive. We have added some content about the relationship between red meat, processed meat and risk of diabetes in the discussion section.

Reviewer 2 Report
The manuscript “Trajectories of meat intake and risk of Type 2 Diabetes: Findings from the China Health and Nutrition Survey (1997-2018)” by Liu et al aims at exploring long-term (over 21 y) trajectories of meat intake in Chinese adults with regard to T2D, and identifies a U-shaped relationship between the meat intake and the risk of T2D (with higher risk of T2D found in both low and high intake groups). Three main points need to be addressed to further improve the study and present its (inarguably important) conclusions to a broader audience:
1) The data presented in the Table 1 regarding groups 1 and 4 demonstrate the male gender, urban life, high education and high income to highly correlate with the risk of T2D. While authors present the models adjusted for these covariates, it will be helpful to show actual data on smaller size samples (derived from the study pool), comparing specifically nearly identical (in terms of included parameters) participants the differ only in reported meat intake.
2) Several parameters (such as income, education, urban vs rural lifestyle) are dynamic and likely to change over 21+y study. It is important to specify when those were registered, and how the changes in these parameters over the course of study were addressed by the authors.
3) The left branch of the U-shaped curve (increased risk in the low intake group) deserves more explanations in the discussion section.
Author Response
Point 1: The data presented in the Table 1 regarding groups 1 and 4 demonstrate the male gender, urban life, high education and high income to highly correlate with the risk of T2D. While authors present the models adjusted for these covariates, it will be helpful to show actual data on smaller size samples (derived from the study pool), comparing specifically nearly identical (in terms of included parameters) participants the differ only in reported meat intake.
Response 1:
Thank you very much for your constructive suggestions.The number of people in the four groups of meat intake trajectories is uneven, with the results indicating 883 people in the first group and 131 people in the fourth group. The baseline characteristics of the four estimated trajectories show that the distribution of the first and fourth trajectories varies among gender, age,and different groups of education level, income level, etc, which indicates that these factors are also highly related to the risk of T2D. Later, we adjusted many covariates. After adjusting as many covariates as possible, we still observed that the risk of T2D in the “high intake group” (Group 4) was more than twice that of the reference group. We believe that our research results are reliable. Your suggestion for nested case control is very reasonable, which is to match a certain number of samples with the same demographic and sociological characteristics, but only with different meat intake trajectories. This is a scientific and professional suggestion. However, after comprehensive consideration, we have adopted the group-based trajectory modeling that has strict requirements for samples. In order to achieve the implementation of this method, we have implemented strict inclusion and exclusion criteria, resulting in a significant loss of sample size. Considering the loss of sample size and statistical effectiveness, we decided to maintain the original research design and technology roadmap. We will implement your suggestion in future suitable research.
Point 2: Several parameters (such as income, education, urban vs rural lifestyle) are dynamic and likely to change over 21+y study. It is important to specify when those were registered, and how the changes in these parameters over the course of study were addressed by the authors.
Response 2:
We initially used the measurements of parameters when participants first entered the "survival analysis period" (2009 or 2015) as covariate, but we believe your suggestion is reasonable and we have decided to adopt it. We reanalyzed the data and use time-dependent covariates instead of baseline covariates. We assessed covariates between 2009 and 2018, and both baseline and follow-up covariates were used in the present study. With diabetes as the dependent variable and the number of years of follow-up as the time variable, a two-level COX regression analysis was conducted (one level for individuals and two levels for families). The specific revised results have been highlighted in the manuscript.
Point 3: The left branch of the U-shaped curve (increased risk in the low intake group) deserves more
explanations in the discussion section.
Response 3:
According to the current literature, many studies have examined the U-shaped dose-response relationship between meat intake and T2D. However, there has been a lot of discussion on the right branch of the U-shaped curve, and there is not much literature support for the explanation of the left branch. In the future, there should be more research to explain the impact of low meat consumption on T2D, which is also a matter of concern. Thank you for your suggestion. We have decided to include some explanations and speculations about The left branch of the U-shaped curve in the discussion section, although there is currently limited research available on this aspect.
